# Characteristics of Soil Dissolved Organic Matter Structure in Albi-Boric Argosols Profiles Through Straw Incorporation: A Fluorescence Spectroscopy Study

**DOI:** 10.3390/plants14111581

**Published:** 2025-05-22

**Authors:** Baoguo Zhu, Enjun Kuang, Qingying Meng, Haoyuan Feng, Miao Wang, Xingjie Zhong, Zhichun Wang, Lei Qiu, Qingsheng Wang, Zijie Wang

**Affiliations:** 1Jiamusi Branch, Academy of Agricultural Sciences of Heilongjiang/Key Laboratory of Breeding and Cultivation of Main Crops in Sanjiang Plain, Jiamusi 154007, China; zhubaoguo82@163.com (B.Z.); zxj15318315825@163.com (X.Z.);; 2Heilongjiang Academy of Black Soil Conservation and Utilization, Harbin 150086, China; 3Northeast Institute of Geography and Agroecology, Chinese Academy of Sciences, Changchun 130102, China

**Keywords:** albi-boric argosols, straw incorporation, soil dissolved organic carbon, fluorescence spectroscopy, nutrient efficiency

## Abstract

Albi-boric argosols, mainly distributed in the Sanjiang Plain of Heilongjiang Province, China, accounting for over 80% of the total cultivated land area, is characterized by a nutrient-deficient layer beneath black soil. This study addresses the challenges of modern agriculture by investigating the impact of straw incorporation on soil dissolved organic carbon (DOC) and its structures in albi-boric argosols, profiles, using fluorescence excitation–emission spectroscopy and parallel factor analysis (PARAFAC). Three treatments were applied: undisturbed albi-boric argosols (C), mixed albic and illuvium layers (M), and mixed albic and illuvium layers with straw (MS). Results showed that the yield of M and MS increased by 9.9% and 13.0%, respectively. There was a significant increase in DOC content, particularly in the MS treatment. Fluorescence index (FI) values ranged from 1.65 to 1.86, biological index (BIX) values were less than 1, and humification index (HIX) values were below 0.75, indicating a mix of plant and microbial sources for DOC, autochthonous characteristics, and weaker humification degree. PARAFAC identified two/three individual fluorophore moieties that were attributed to fulvic acid substances, soluble microbial products, and tyrosine-like substances, with microbial products as the dominant component. This study demonstrates the effect of improving barrier soil and maintaining sustainable agriculture by enhancing soil quality.

## 1. Introduction

Soil dissolved organic carbon (DOC) plays an integral role in the soil’s nutrient pool. As the most dynamic carbon component, it is not only the main energy and material source for soil microorganisms [1] but also plays an important role in soil carbon cycling due to its easy decomposition and rapid turnover [2]. It is an important indicator of soil environmental health and changes in soil quality [3]. The content and structural characteristics of DOC are significantly affected by tillage methods, organic matter application [4], soil environment, and other conditions [5].

Albi-boric argosols, one of the significant agricultural soils in northeastern China, is predominantly found in Heilongjiang and Jilin provinces. Characterized by a typical gray-white layer (Aw layer) beneath the black soil layer, it contains a substantial amount of SiO_2_ powder and iron–manganese nodules in its lower layer [6]. The albi-boric argosols in Northeast China is comparable to “Lessivage” in France, “Pseudo gley” in Germany, “pseudo podzolic” in Russia, and some clay soils in the United States [7]. In Heilongjiang Province, 80% of the country’s grain is produced here, making it an important commodity grain base. The total area of albi-boric argosols in our province is approximately 3.312 million hectares, with 884,000 hectares cultivated, accounting for 25.4% of the province’s total cultivated land area [8]. Albi-boric argosols is known for its shallow black soil layers, compact and complex albic horizons, nutrient depletion, poor water permeability, and aeration, which lead to frequent drought and waterlogging disasters [9], ultimately resulting in low crop yields. Improving and properly utilizing these soil resources is crucial for transforming the low-yield situation in albi-boric argosols regions and reinforcing the role of black soil as the “ballast stone” of grain production.

Applying a self-developed straw-incorporated subsoiling plow can significantly enhance soil physical properties by ensuring the lower albic horizons and illuvium layers are mixed in the same proportion (1:0.5–1) while maintaining the black soil layer’s position. Additionally, incorporating materials such as straw, organic fertilizers, and phosphorus fertilizers into the mixing soil layer during the soil improvement process addresses nutrient deficiency [10,11,12,13]. This technique has shown remarkable results in state-owned and local cities and counties in the Sanjiang Plain.

Organic fertilizer and straw improve the content of soil DOC, enhance the decomposition and metabolism of soil microorganisms, and, at the same time, increase and simplify the structure of the fulvic acid component [4]. Incorporating straw into soil increases the relatively simple structured fulvic acid components, simplifying the DOC structure [14]. Straw incorporation also impacts the organic carbon pool below a 40 cm depth [15]. After deep tillage with the straw return, the fluorescence index analysis indicated that black soil areas were affected by both internal and external influences, which had the lowest autogenic characteristics, were greatly affected by exogenous input, had the highest humification coefficient, and had a more stable DOC structure [16].

Northeastern China is an important grain production base, and the adverse effects of obstacles pose a threat to grain production and the ecological environment in the albi-boric argosols. Therefore, increasing the organic matter content of albi-boric argosols and improving soil fertility play an important role in agricultural production. Most studies focus on increasing SOC content through the application of organic materials to increase soil carbon pool, significantly affecting the accumulation of deep organic carbon pools in black soil, red soil, and paddy soil [17], but there have been fewer reports on the effect of applying organic materials on the structure of DOC components in albi-boric argosols. In this review, we hypothesized chemical and optical properties of DOC presenting distinct relationships with straw incorporation and DOC fractions. The main objectives of this review were to (1) assess whether the DOC content in mixing layers of albi-boric argosols had little changes and (2) the higher content with straw incorporation and varied DOC fractions of fluorescent materials. This study aims to explore the changes in the content and components of DOC following straw incorporation into the mixing layer, providing a theoretical foundation for understanding the dynamic changes in DOC and its constituent substances in albi-boric argosols.

## 2. Materials and Methods

### 2.1. Experimental Site

The site is located in Heilongjiang Province, Beidahuang Group Farm “854“(45°58~46°10′ N, 132°46′~133°15′ E), which is situated in the eastern part of the Sanjiang Plain (Figure 1), at the southern foot of the Wanda Mountains, and in the northeastern part of the Xingkai Lake sedimentary plain of the Muleng River. The spring and autumn seasons are short and characterized by variable weather conditions. The average annual temperature is 2.4 °C and the active accumulated temperature above 10 °C is 2442.8 °C per year; the frost-free period lasts for 131 days. The soil type at the experimental site is albi-boric argosols, and the soil profile is distributed as follows: a 0–20 cm layer of black soil (topsoil), a 20–40 cm layer of albic, and a 40–60 cm layer of illuvium.

### 2.2. Experimental Design

The amelioration of albi-boric argosols by mechanics was carried out on Farm 854 every 5 years from 2012. When performing mechanical operations, we applied straw to the mixed soil layers in the autumn of 2022, and soil samples were collected before the harvest in the autumn of 2023.

Based on the binary texture characteristics of albi-boric argosols, this study established the soil amendment principle of “20 cm topsoil inversion combined with 30–40 cm subsoil mixing”. Building upon this theoretical framework, Araya et al. [18] and Liu et al. [19,20,21] successfully developed a three-section subsoil layer-mixing plow. This innovative implement maintains the original position of the surface humus layer while uniformly mixing the albic horizon and illuvial horizon at a 1:1 thickness ratio, thus resulting in significant improvements in soil permeability and water storage capacity. Notably, its tillage depth exceeds 50 cm. To address the inherent nutrient deficiency in albi-boric argosols’ subsoil, our research team further optimized this technology by developing a patented straw–subsoil mixing plow (Patent No.: ZL201420220290.6). This enhanced implement achieves deep-layer incorporation of organic amendments like straw into albi-boric argosols subsoil, demonstrating remarkable field efficacy in soil improvement. The details of the soil improvement mechanism can be found in Figure 2 and the mechanical references in our team’s previous work [22]. 

The plough system comprises four sequentially arranged components mounted on the frame: (1) primary ridge-rolling plow, (2) secondary root-stubble scraper, (3) tertiary bar-type subsoil breaker, and (4) quaternary bar-type stubble-subsoil mixer. During operation, the primary plow inverts the 20 cm topsoil layer. The secondary implement subsequently scrapes 3–5 cm of surface-level straw and root residues into the furrow created by the primary plow. The tertiary unit vertically fractures approximately 20 cm of subsoil along the primary furrow. The quaternary module simultaneously engages the tertiary furrow to excavate an additional 20 cm subsoil layer. During subsequent passes, the processed 20 cm topsoil layer (now stripped of surface residues) becomes inverted over the ameliorated subsoil mixture, completing the cyclic soil restructuring process.

The trial field implemented a maize–soybean rotation system. During maize seasons, straw incorporation using a John Deere S760 combine harvest (339HP) with an integrated shredding system, producing 2~10 cm fragments.

The experiment consisted of three treatments arranged in a randomized block design with three replications. Each plot had an area of 0.1 hectares. The specific treatments were as follows. Treatment 1: albi-boric argosols control (C). The original layers of the albi-boric argosols, from top to bottom, were black soil layer (Ap, 0–20 cm), albic layer (Aw, 20–40 cm), and illuvial layer (B, 40–60 cm). Treatment 2: mixed Aw and B layer except topsoil (M). The Ap layer in the albi-boric argosols was moved away, Aw and B layers were mixed, and then Ap was put back in place. Treatment 3: mixed Aw+B layers with maize straw incorporation (MS). Based on treatment 2, finely crushed maize straw was integrated into the Aw+B layer at a depth of 20–40 cm at an amount of 20 t·hm^−2^. At the same time, urea application of 408 kg/hm^2^ was used to adjust the C:N ratio, co-incorporated with shredded straw. The straw was removed from the plots without applying it.

Soybeans were cultivated as the primary crop in all three treatment plots. Fertilizer management included the application of urea (containing 46% nitrogen) at a rate of 30 kg per hectare, diammonium phosphate (comprising 18% nitrogen and 46% P_2_O_5_) at 30 kg per hectare, and potassium chloride (with 60% K_2_O) at 60 kg per hectare. This fertilizer blend was applied in a single application during the spring season. All other management measures remained consistent after soil treatment.

A systematic soil sampling procedure was conducted post-harvest. The collection of soil samples involves manually excavating a soil profile of 50 cm (L) × 50 cm (W) × 70 cm (D) and taking samples in layers of 0–20 cm, 20–40cm, and 40–60 cm. For each experimental treatment, triplicate georeferenced sampling points are systematically established, with corresponding stratigraphic specimens homogenized to form composite samples per horizon, ensuring triplicate biological replicates per treatment group. The collected samples underwent thorough processing to eliminate any remaining straw or root material. Subsequently, the samples were air-dried and sieved through a 2 mm mesh and stored in a sealed bag at room temperature to facilitate the analysis of DOC and fluorescence structures. The table below presents a comprehensive overview of the general physicochemical properties observed in the soil samples from the three treatment plots (Table 1). The experimental area features flat terrain and lacks irrigation facilities.

### 2.3. Measurement Methods

#### 2.3.1. Determination of Soil Physicochemical Indicators

Soil chemical properties: available P was measured by the NaHCO_3_ extraction–Mo-Sb colorimetry method, and the cation exchange capacity (CEC) was determined by the ammonium acetate method. Soil organic carbon (SOC): the air-dried soil sample was weighed at 0.0100 g and subjected to acid digestion using a 2 mol·L^−1^ hydrochloric acid solution. Subsequently, the resulting solution was filtered through a 0.45 μm membrane, and the soil’s organic carbon content was assessed using a total organic carbon analyzer (Multi N/C 3100, Analytik Jena, Germany) [23].

#### 2.3.2. Fluorescent Sample Handling and Measurement Methods

DOC: a 3 g portion of air-dried soil was mixed with 30 mL of ultrapure water. The mixture was horizontally oscillated at a speed of 200 r·min^−1^ for 24 h at room temperature. After centrifugation at 12,000 r·min^−1^ for 20 min, the supernatant was filtered through a 0.45 μm membrane before being analyzed using a total organic carbon analyzer (Multi N/C 3100, TOC instrument) to determine the concentration of DOC [24].

Fluorescence spectrum: a small amount of solution was taken for each treatment and ultrapure water was added according to its concentration, adjusting the DOC concentration to a uniform concentration of 10 mg·L^−1^ to eliminate the impact of solution concentration of DOC. Subsequently, the three-dimensional fluorescence spectrum was measured using a fluorescence spectrometer (Hitachi F-7000, Tokyo, Japan) with a scanning range of 200 to 600 nm for both excitation wavelength (Ex) and emission wavelength (Em), utilizing a bandwidth of 10 nm and a scanning speed of 1200 nm·min^−1^, using ultrapure water as blank.

Fluorescence index (FI, f450/500) refers to the ratio of the emission wavelength at 450 (Em = 450) nm to the excitation wavelength at 500 (Em = 500) nm when the excitation wavelength is 370 (Ex = 370) nm [25]. These indicators are used to identify the sources of humus in DOC. It suggests that there are generally two sources of DOC, plant sources and microbial sources, with FI values ranging between 1.4 and 1.9. An FI < 1.4 indicates that DOC is mainly composed of external substances like plant litter and root exudate decomposition. An FI > 1.9 indicates that DOC mainly comes from the metabolism and degradation of self-generated soil microorganisms. An FI falling between 1.4 and 1.9 indicates that DOC contributes a mixture of plant and microbial sources [26].

The biological Index (BIX) refers to the ratio of Em = 380 to Em = 430 nm when Ex = 310 nm [16]. Typically, 0.6 < BIX < 0.7, indicating a minor contribution of endogenous sources in DOC components. A BIX range of 0.7 to 0.8 suggests moderate recent autochthonous features, while a BIX range of 0.8 to 1.0 indicates a significant contribution from DOC components. When BIX > 1.0, the components are mainly autochthonous and the organic matter is newly generated [26].

The humification index (HIX) is defined as the ratio of the integrated values at emission wavelengths of 435–480 nm to the sum of integrated values at emission wavelengths of 435–480 nm and 300–345 nm when the excitation wavelength is 254 nm. HIX is primarily utilized to assess the level of humification of DOC, with a higher degree of humification [27]. The PARAFAC in our study was performed in MATLAB using the DOMFlour toolbox for MATLAB [28]. The influence of Raman scattering on fluorescence data was eliminated when using PARAFAC.

### 2.4. Statistical Analysis

The statistical analysis of the data was conducted using Excel 2010 and SPSS 22.0 software. One-way ANOVA and Tukey’s HSD method were used for the analysis of variance and multiple comparisons (α = 0.05). The three-dimensional fluorescence spectra were plotted and parallel factor analysis was conducted using Matlab 2013 software. The area integration of fluorescence spectral indexes was performed using Origin 2021 software.

## 3. Results

### 3.1. Soybean Yield Factors and Soil Chemical Properties

The M and MS treatments increased soybean plant weight, pod numbers, seed numbers, 100-grain weight, and soybean yield (Table 2). Compared with the C treatment, the yield of M and MS increased by 9.9% and 13.0%. The MS treatment had a better effect on soybean yield and factors. The application of straw significantly increased available P and K content compared with no application, as shown in Table 3.

### 3.2. Variation in Soil DOC Content

DOC shows robust activity but is also susceptible to loss, influenced by the addition of organic materials like straw [12]. In the undisturbed surface horizon soil (treatment C), the DOC content decreased gradually with increasing soil depth. In the 0–10 cm soil layer, the DOC content varied from 250 to 280 mg·kg^−1^, while, in the 50–60 cm soil layer, the DOC content was consistent across all treatments, ranging from 120 to 136 mg·kg^−1^. Following the application of straw, the mixed soil layer in treatment M showed a significant increase in DOC content compared to the undisturbed horizon of treatment C, with an increase ranging from 32.4% to 66.29%. This led to a notable rise in soil DOC content.

The ratio of dissolved organic carbon to soil organic carbon (DOC/SOC) is commonly used to characterize the activity of SOC and its sensitivity to agricultural management practices. In the undisturbed soil of the surface horizon, DOC/SOC was consistently higher compared to other treatments throughout the soil profile, except for the 40–50 cm soil layer, where treatment MS exhibited the highest DOC/SOC (Figure 3).

### 3.3. Analysis of Soil DOC Component Fluorescence Index

As shown in Table 4, the FI for all treatments ranged from 1.65 to 1.86, indicating a distinct signature of a blend of organic compounds originating from plants and microorganisms. The FI values for the black soil layer ranged from 1.65 to 1.72, while the FI values for the albic layer/mixed soil layer in each treatment ranged from 1.78 to 1.86. Furthermore, the FI values for the illuvium layer in each treatment ranged from 1.75 to 1.86.

The sequence of BIX values in the undisturbed albi-boric argosols followed the order of illuvium layer > albic layer > black soil layer. Only the BIX value for the black soil layer was less than 1, indicating its inherent characteristics of self-origin. This aligns with the inherent traits of the black soil layer. The BIX values for the albic and illuvium layers were greater than 1, primarily influenced by biological activity. However, following the application of straw in the mixed soil layer, the BIX value decreased.

The value of HIX in all treatments showed higher values in the Ap layer and lower values in the A/Aw+B layers. In the black soil layer, the HIX values ranged from 0.66 to 0.75, falling within the weak humification range. The addition of straw to the mixed layer stimulated the soil DOC, enhancing its instability. The HIX values for the illuvium layer in each treatment ranged from 0.41 to 0.56, with treatment M exhibiting the highest value.

### 3.4. The Fluorescent Spectral Characteristics of Soil DOC Components

By conducting parallel factor analysis on the fluorescent components of soil DOC in various treatments and soil layers, two substance categories were identified in the black soil layer of treatment C (Figure 4). These categories were humic-acid-like substances (250–300 nm/400–460 nm) [29,30] and soluble microbial metabolites (280–300 nm/350–370 nm). In contrast, the other treatments resulted in three fluorescent components: humic-acid-like substances (240 nm–280 nm/400–460 nm), soluble microbial metabolites (280 nm-300 nm/340–370 nm) [31], and tyrosine-like proteins (<220nm/300–330 nm) [32]. The humic-acid-like substances in the UV region exhibit one excitation peak and one emission peak, known as Peak A. This peak represents organic matter with low degradability and considerable molecular weight, reflecting the soil’s capacity to supply and buffer fertility. The soluble microbial metabolites, characterized by Peak T, also exhibit one excitation peak and one emission peak. Peak T mainly corresponds to the metabolic products derived from bacterial and microbial degradation, which can be bound or free within proteins. The tyrosine-like proteins of the C3 component exhibit one excitation peak and one emission peak, known as Peak B. The fluorescence peaks of humic-acid-like components were located at long excitation and emission wavelengths, indicating that the reorganized components contain organic substances with high molecular weight and high aromaticity, which are difficult to decompose and utilize.

### 3.5. F_max_ Analysis of Soil DOC

After the soil improvement treatments, significant variations in the total fluorescence intensity of soil DOC were observed across different soil layers. The black soil layer of treatments C, M, and MS exhibited no significant changes in F_max_. However, the albic layer and illuvium layer of treatments M and MS exhibited a decreasing trend compared to treatment C. Specifically, the mixed soil layer of treatments M and MS showed a decrease of 10.2% and 30.8%, respectively, compared to the albic layer at the same depth in treatment C. Additionally, the illuvium layer of treatments M and MS decreased by 23.5% and 16.2%, respectively, compared to the illuvium layer in treatment C (Figure 5).

Treatment C’s black soil layer contained two substances, with soluble microbial metabolites accounting for over half of the proportion. In the albic layer of treatment C, the proportion of soluble microbial metabolites was 55.1%, followed by humic-acid-like and tyrosine-like proteins, accounting for 20.2% and 24.7%, respectively. In treatment M’s mixed soil layer, the proportion of soluble microbial metabolites was 56.5%, showing a decrease in humic-acid-like substances to 17.1%. In comparison, tyrosine-like proteins slightly increased to 26.4%. In the mixed soil layer of treatment MS, the proportion of soluble microbial metabolites was 51.3%, showing a decrease in humic-acid-like substances to 19.6%, while tyrosine-like proteins slightly increased to 29.1%.

Regarding the illuvium layer, treatment C had a proportion of 56.5% for soluble microbial metabolites, with humic-acid-like and tyrosine-like proteins accounting for 15.4% and 28.1%, respectively. In treatment M, there was a slight decrease in the proportion of soluble microbial metabolites to 47.4%, with humic-acid-like and tyrosine-like proteins accounting for 27.0% and 25.6%, respectively. As for treatment MS, the proportion of dissolved organic matter in the illuvium layer was 55.8%, with humic-acid-like and tyrosine-like proteins making up 20.8% and 23.4%, respectively (Figure 5).

### 3.6. The Correlation Analysis of Soil SOC and DOC Content, as Well as Their Components

A significance analysis of DOC and its fluorescent components among different treatments was conducted (Table 5). The soil DOC content showed a significantly positive correlation with SOC (*p* < 0.01). Moreover, there was a highly significant negative correlation (*p* < 0.01) between tyrosine-like proteins and soluble microbial metabolites and the ratio of DOC to SOC.

## 4. Discussion

### 4.1. Influence of Different Treatments on Soil DOC Content

Soil DOC primarily originates from plant residues, external organic inputs, microbial metabolites, and soil organic matter decomposition. Changes in cultivation methods, fertilization practices, or climate factors can lead to corresponding alterations in the chemical components of soil DOC [16]. Furthermore, soil DOC content significantly correlates with SOC content [33]. In the study site, which comprises components of a black soil layer, an albic layer, and an illuvium layer, the top black soil layer exhibits higher nutrient and organic matter content. However, the albic layer, with a clay-to-silt ratio of 3:1 [34], suffers from soil compaction, poor permeability, and high soil hardness, making it difficult for roots to penetrate. By mixing the albic layer with the illuvium layer, the clay-to-silt ratio of the albic layer was adjusted, thereby improving its unfavorable physical properties [8]. Based on this experiment, it can be observed that the soil DOC content in the mixed soil layer increased compared to the undisturbed albic layer, indicating that improved physical properties contribute to enhancing soil fertility and nutrient availability. In the 50–60 cm soil layer, the DOC content of all treatments was low and similar, ranging from 10.5 to 14.5 mg·kg^−1^. This can be attributed to the illuvium layer, which contains a high percentage of clay particles (56.5%) [34], leading to the increased adsorption of DOC and a weakened decomposition capacity of soil microorganisms towards organic matter. After the addition of straw (MS treatment), the soil DOC content in the mixed soil layer significantly increased. This is because straw contains a substantial amount of carbon sources, providing favorable substances and energy for microbial growth. This, in turn, stimulates the decomposition of easily decomposable soil organic matter and facilitates the transformation of soil organic carbon into more active organic carbon fraction [35].

### 4.2. Influence of Different Treatments on Soil DOC Component Fluorescence Indexes

Fluorescence spectral indexes (FI, BIX, and HIX) are commonly used to characterize the structural properties of soil humus [26]. Among them, FI reflects the origin of humus. An FI value below 1.4 indicates the presence of terrestrial organic matter from external inputs or microbial metabolites, while an FI value above 1.9 suggests the presence of internally derived microbial degradation products. An FI value between 1.4 and 1.9 indicates a significant influence from both sources. In this experiment, the FI values of the black soil layer in the plowed soil ranged from 1.65 to 1.72, while the FI values of the albic layer/mixed soil layer treatments ranged from 1.78 to 1.86, and the FI values of the illuvium layer treatments ranged from 1.75 to 1.86. The FI values of all treatments fell within the range of 1.4 to 1.9, indicating a mixed composition of soil DOC originating from both plants and microorganisms. The variation in FI values after straw incorporation ranged from 1.4 to 1.8, which is consistent with previous research findings [26,36].

BIX is used to measure the contribution of autochthonous organic matter [26]. When BIX values range from 0.6 to 0.7, 0.7 to 0.8, and 0.8 to 1.0, they represent a minor, moderate, and strong autochthonous origin of dissolved organic matter, respectively. A BIX value greater than 1 indicates materials generated through biological activities. In this experiment, only the BIX value of the black soil layer was less than 1, indicating its strong autochthonous characteristics. On the other hand, the albic layer and illuvium layer had BIX values greater than 1, indicating a less significant autochthonous origin in the albic layer and relying more on biological activities. The decrease in microbial activity following straw incorporation could be attributed to the high energy demand for the straw’s decomposition, which competes for nutrients with microbial activity.

HIX represents the degree of humification of soil organic matter [27]. It is positively correlated with the degree of humification of soil DOC, where higher HIX values indicate better stability. In this study, the HIX values of all treatments were less than 0.75. The HIX value of the black soil layer was higher than that of other layers, ranging from 0.66 to 0.75, which was still within the range of weak humification. The degree of humification decreased in the albic and illuvium layers after mixing but showed a slight increase after straw incorporation. The incorporation of straw activated the soil DOC, increasing its instability. As a novel carbon source, straw not only enhanced the carbon content but also activated soil organic carbon. The unique soil composition of black soil provides favorable conditions for humus accumulation. This explains why the black soil layer has significantly higher organic matter content than other soil types. Research has shown that the HIX values in brown soil and red soil range from 0.38 to 0.46 [36].

### 4.3. The Influence of Different Treatments on the DOC Components

Despite no significant changes in DOC content, the influence on its components was evident [37]. Different feedback on straw application is observed due to the different soil types. Studies have shown that combined application of organic and inorganic fertilizers in black soil, red soil, paddy soil, and fluvo-aquic soil can enhance both DOC and total organic carbon content [38], though differences exist between soil types. Red soil exhibits higher proportions of microbially derived components [39], black soil shows varying degrees of increase in amine substances [35], while combined application of chemical fertilizers with straw returning in paddy soil leads to more complex and stable aromatic compound structures [40]. Different fertilization materials, straw return methods, and tillage practices have been found to increase the content of relatively simple-structured fulvic acid components in DOC while simplifying their structural complexity [12,41].

This experiment, conducted using parallel factor analysis, revealed three fluorescence components: humic-acid-like substances, soluble microbial metabolites, and tyrosine-like proteins. Soluble microbial metabolites accounted for over 50% of the total, which is consistent with the analysis of fluorescence spectral indexes conducted earlier, indicating the predominant role of microbial activity. Humic-acid-like substances are associated with hydroxyl and carboxyl groups in DOC and are generally of exogenous origin [42]. Some studies have showed that the application of organic fertilizer increases the metabolic products of soluble microorganisms in DOC, leading to an increase in fulvic acid and humic acid substances and a simplification of their structures [12]. Straw incorporation into deep black soil can increase fulvic-acid-like substances and decrease tryptophan-like substances, with no soluble microbial metabolites [36]. Other researchers showed the dominance of fulvic-acid-like substances in black deep-layer soil and that of main tryptophan-like substances in black thin-layer soil [16].

In this experiment, the addition of external carbon had a minimal impact on soil DOC. Tyrosine-like proteins, which are smaller and more easily degradable than tryptophan-like substances, experienced a decrease in total fluorescence intensity after straw incorporation. However, all three substances exhibited an increasing trend, with an increase of 10.2% in tyrosine-like proteins. This suggests that incorporating straw transformed DOC into more superficial molecular structures. Metabolites of dissolved microorganisms in soil are generally proportional to the metabolic rates and quantity of soil microorganisms [43]. Studies have shown that the proportion of dissolved microbial metabolites in high-yield soils in DOC is relatively high. As a low-yield soil, albi-boric argosols also has a relatively high proportion of dissolved metabolites, likely due to increased microbial metabolism. Proteinoids, such as tryptophan and tyrosine, are essential carbon sources for soil microorganisms and significant components of soil amino acids [44]. Microbial metabolism gradually results in the accumulation of tyrosine-like proteinoids and tryptophan-like proteinoids in the soil.

Studies have shown that, in low-carbon soils, there are three fluorescence peaks: a protein-like fluorescence peak, a fulvic-acid-like fluorescence peak, and a humic-acid-like fluorescence peak. However, in high-carbon soils, only the fulvic-acid-like and humic-acid-like fluorescence peaks are present [45]. In this experiment, various types of substances that may be associated with the soil type were identified. The conclusion mentioned above was drawn from red soil, while, in meadow soil and black soil, fulvic acid and proteins similar to tyrosine also prevail [46]. The excitation/emission wavelength of the fluorescence peak in the three-dimensional spectrum increases, which is called redshift. It stands for the materials being relatively complex and the degree of humification being high. An increase in the excitation/emission wavelengths of fluorescence peaks in EEM spectra, termed a “redshift”, signifies enhanced structural complexity and a higher degree of humification in organic materials [47]. After incorporating straw into the soil, there is a tendency for the relatively simple structure of fulvic-acid-like substances to become even simpler [12].

## 5. Conclusions

Ameliorating the barrier layer of albi-boric argosols had a positive impact on soil DOC content and crop yield with M and MS treatments, especially when straw was incorporated. The Ap layer consistently exhibited higher DOC content than the Aw and B layers. Mixing soil layers and straw application improved soil DOC content and the DOC/SOC ratio. The Ap layer displayed strong endogenous characteristics and a high humification coefficient, making it more conducive to structural stability in albi-boric argosols. After the application of straw in this experimental area, the excitation wavelength and emission wavelength of humic-acid-like substances decreased, and there was a trend of humic-acid-like substances transforming into fulvic-acid-like substances, simplifying the structure; the protein substances were also increased, indicating that straw application enhances the capacity of soil’s fertilizer supply, thereby increasing crop yield. This study highlights the potential of straw incorporation as an effective strategy for improving albic soil quality, aligning with innovative nutrient management practices to enhance crop yield, quality, and nutrient efficiency, thus addressing the challenges of sustainable agriculture.

## Figures and Tables

**Figure 1 plants-14-01581-f001:**
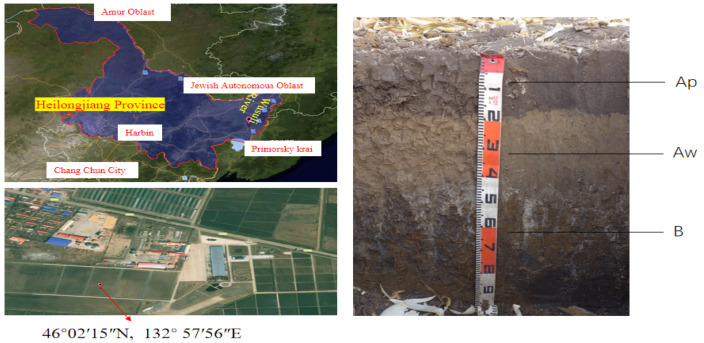
The location of the experiment site and the profile of albi-boric argosols in Farm “854”.

**Figure 2 plants-14-01581-f002:**
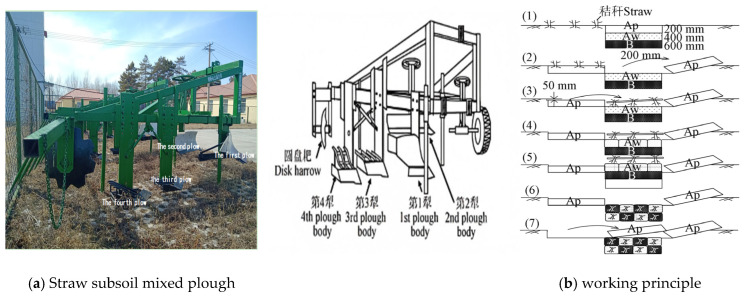
Improvement mechanism of albi-boric argosols. (**a**) Straw subsoil mixed plough; (**b**): working principle (Ap: top soils; Aw: albic horizon; B: illuvial horizon. (1)–(7) are the working steps of the plough.)

**Figure 3 plants-14-01581-f003:**
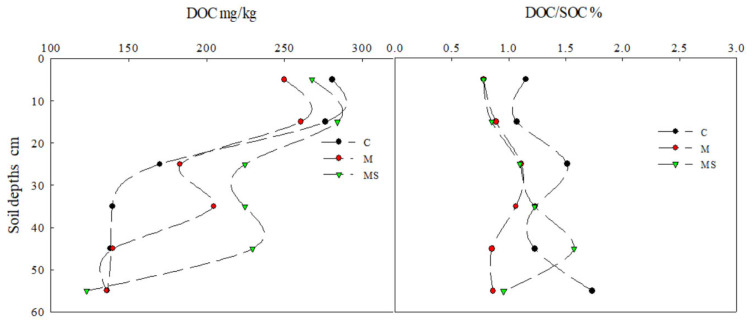
The content of DOC and the DOC/SOC rate under different treatments.

**Figure 4 plants-14-01581-f004:**
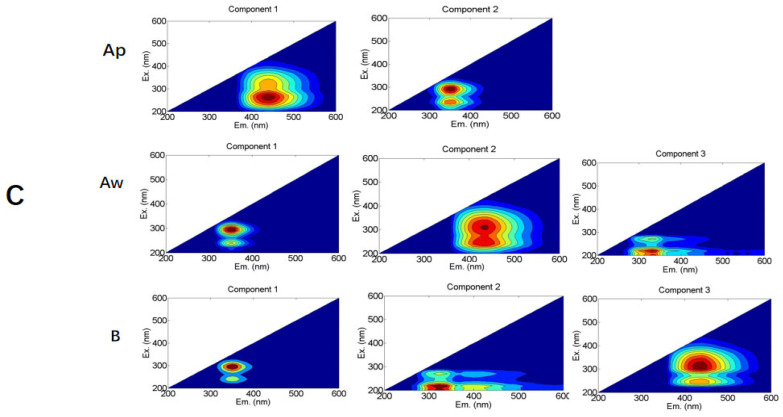
Three-dimensional fluorescence components of soil DOC in different treatments.

**Figure 5 plants-14-01581-f005:**
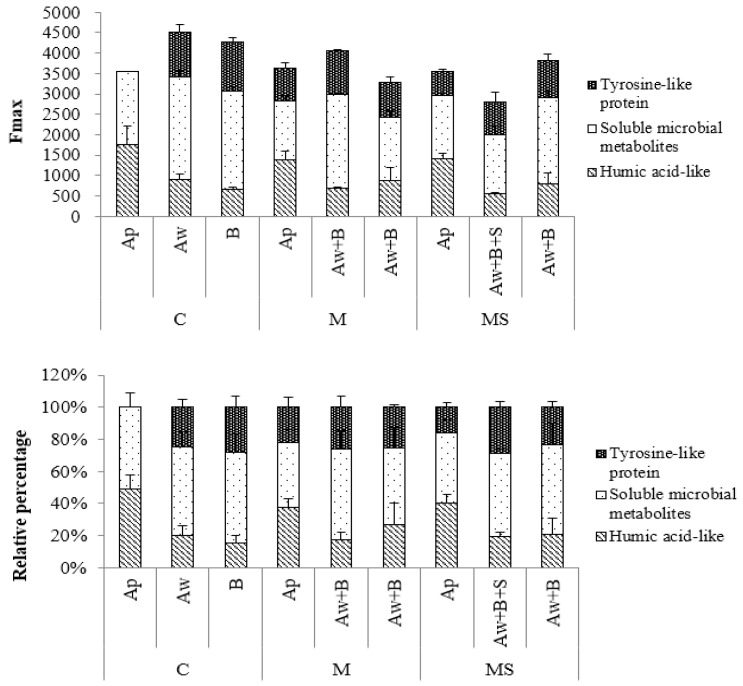
F_max_ and relative percentage of soil DOC fluorescence components.

**Table 1 plants-14-01581-t001:** Physicochemical properties of soil samples.

Layers	Bulk Density (g·cm^−3^)	org-C (g·kg^−1^)	TN (g·kg^−1^)	C/N	CEC (cmol·kg^−1^)	Available Phosphate (mg·kg^−1^)
Ap (0–20 cm)	1.18 ± 0.04 c	13.84 ± 0.22 a	1.17 ± 0.07 a	14:1	19.3 ± 0.37 b	21.8 ± 0.29 a
Aw (20–40 cm)	1.55 ± 0.04 a	3.77 ± 0.08 c	0.83 ± 0.02 b	5:1	14.5 ± 0.23 c	6.0 ± 0.20 c
B (40–60 cm)	1.43 ± 0.03 b	4.58 ± 0.06 b	0.69 ± 0.02 c	7:1	31.3 ± 0.56 a	9.2 ± 0.17 b

Note: different lowercase letters in the same column indicate significant differences at the 0.05 level.

**Table 2 plants-14-01581-t002:** Soybean yield changes under different treatments.

Treatment	Plant Height (cm)	Number of Pods per Plant	Number of Seeds per Plant	100-Grain Weight (g)	Yield (kg hm^2^)	Increasing Yield %
C	90.40 ± 0.61 c	28.40 ± 0.26 b	58.30 ± 0.75 c	18.57 ± 0.19 a	2598.8 ± 83.5 a	—
M	94.54 ± 0.69 b	30.34 ± 0.23 a	63.93 ± 0.63 b	18.61 ± 0.11 a	2855.5 ± 150.2 a	9.88
MS	97.20 ± 0.76 a	31.20 ± 0.58 a	65.70 ± 0.58 a	18.62 ± 0.12 a	2936.5 ± 208.4 a	13.00

Note: different lowercase letters in the same column indicate significant differences at the 0.05 level.

**Table 3 plants-14-01581-t003:** Soil chemical properties in different treatments.

Treatment	Soil Layer (cm)	Available N (mg·kg^−1^)	Available P (mg·kg^−1^)	Available K (mg·kg^−1^)	pH
C	Ap	131 ± 6.08	17.7 ± 0.62	162 ± 5.57	5.45 ± 0.19
	Aw	78 ± 5.57	5.4 ± 0.35	121 ± 4.58	5.60 ± 0.08
	B	68 ± 3.61	9.7 ± 0.44	172 ± 6.24	5.90 ± 0.11
M	Ap	136 ± 8.54	17.8 ± 0.62	164 ± 6.00	5.46 ± 0.16
	Aw+B	85 ± 4.00	7.5 ± 0.32	137 ± 7.81	5.68 ± 0.07
	Aw+B	71 ± 3.61	9.8 ± 0.57	170 ± 6.24	5.94 ± 0.07
MS	Ap	126 ± 9.17	16.1 ± 0.60	160 ± 6.56	5.45 ± 0.09
	Aw+B+S	96 ± 7.94	13.6 ± 0.36	155 ± 5.29	5.69 ± 0.09
	Aw+B	79 ± 5.57	10.1 ± 0.33	172 ± 7.00	5.94 ± 0.13

**Table 4 plants-14-01581-t004:** Mean fluorescence spectral indices of soil DOC under different treatments.

	C			M			MS		
Item	Ap	Aw	B	Ap	Aw+B	Aw+B	Ap	Aw+B+S	Aw+B
FI	1.67 ± 0.02 c	1.78 ± 0.05 b	1.86 ± 0.03 a	1.72 ± 0.03 b	1.86 ± 0.07 a	1.75 ± 0.03 b	1.65 ± 0.03 c	1.85 ± 0.02 a	1.79 ± 0.03 b
BIX	0.93 ± 0.13 c	1.40 ± 0.34 b	1.70 ± 0.43 a	0.94 ± 0.16 c	1.67 ± 0.48 a	1.27 ± 0.06 b	0.81 ± 0.04 c	1.38 ± 0.09 a	1.54 ± 0.56 a
HIX	0.71 ± 0.04 a	0.51 ± 0.06 b	0.41 ± 0.05 bc	0.66 ± 0.03 a	0.41 ± 0.07 c	0.56 ± 0.03 b	0.75 ± 0.01 a	0.49 ± 0.03 b	0.46 ± 0.13 b

Note: the values are mean ± standard deviation. Different lowercase letters represent very significant difference (*p* < 0 05).

**Table 5 plants-14-01581-t005:** Pearson correlation analysis between soil DOC content and various fluorescence intensities.

Treatments	DOC	SOC	C1	C2	C3	DOC/SOC
DOC	1.000					
SOC	0.857 **	1.000				
C1	−0.292	−0.111	1.000			
C2	−0.182	−0.313	−0.161	1.000		
C3	0.023	0.304	0.395	−0.698 **	1.000	
DOC/SOC	−0.294	−0.694 **	−0.207	0.385	−0.609 **	1.000

Note: ** significant correlation at the 0.01 level (bilateral). C1 represents humic-acid-like substances, C2 represents soluble microbial metabolites, and C3 represents tyrosine-like proteins. (n = 18).

## Data Availability

The data generated during and/or analyzed during the current study are available from the corresponding author.

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
