# Peer review of "Characteristics of Soil Dissolved Organic Matter Structure in Albi-Boric Argosols Profiles Through Straw Incorporation: A Fluorescence Spectroscopy Study"

_plants, 2025, doi:10.3390/plants14111581_

Round 1
Reviewer 1 Report
Comments and Suggestions for Authors
The authors of the article present the results of research on an interesting field experiment to improve the properties of plantasols by adding plant material (straw) and its effect on the composition of water-soluble organic matter.
The introduction describes the tasks set quite fully and gives an idea of ​​the soils that are the object of the study.
- It is worth noting, however, that the analogies of the plantasol soil type with the classifications of other countries have apparently lost their relevance and need to be verified. For example, in the classification of Russian soils, plantasols correspond to solods, and pseudo-podzolic soils are an outdated concept (line 46)
-As for the structure of the article, section 4 "materials and methods", including table 5 (that is, the entire section) should be moved to the beginning of the article and placed immediately after the introduction and before the presentation of the results obtained.
-The discussion subsection should be supplemented with graphs or figures illustrating the discussion
Author Response
Comment1:It is worth noting, however, that the analogies of the planosol soil type with the classifications of other countries have apparently lost their relevance and need to be verified. For example, in the classification of Russian soils, planosol correspond to solods, and pseudo-podzolic soils are an outdated concept (line 46)
Response1: Thank you for your positive comments and valuable suggestions to improve the quality of our manuscript. Yes, you are right, planosol was an outdated concept. The commonly used vocabulary is Albi Boric Argosols. We revised all the “planosol” to “Albi-Boric Argosols”.
Comment2:-As for the structure of the article, section 4 "materials and methods", including table 5 (that is, the entire section) should be moved to the beginning of the article and placed immediately after the introduction and before the presentation of the results obtained.
Response2: Accept. We sincerely appreciate the constructive suggestion regarding the structural organization of the manuscript. We fully agree that relocating the “Materials and Methods” section to the follow the introduction will enhance the logical flow of paper and align it with the conventional structure of scientific articles.
The original Section 4 ("Materials and Methods") and its associated content (including Table 5) have been moved to Section 2, immediately following the Introduction. Subsequent sections (Results, Discussion, etc.) have been renumbered accordingly. All in-text references to the section and table and graph numbering have been updated to ensure consistency.
Comment3:The discussion subsection should be supplemented with graphs or figures illustrating the discussion
Response3: We sincerely thank the reviewer for this valuable suggestion. But I'm very sorry that I didn't understand the specific meaning of this suggestion. Should we use other research figures in the discussion or annotate the chart we mentioned earlier? Could you show me the details? Thanks again.
Reviewer 2 Report
Comments and Suggestions for Authors
• What is the main question addressed by the research? This article is about Characteristics of Water soluble Organic Matter Structure in Planosol Profiles through Straw Incorporation: A Fluorescence Spectroscopy Study. This study demonstrates the effect of improving barrier soil and maintaining sustainable agriculture by enhancing soil quality, due to the addition of straw.
• Do you consider the topic original or relevant to the field? Does it address a specific gap in the field? Please also explain why this is/ is not the case. Yes, the topi cis original, but lanosol is characterized by a nutrient-deficient layer beneath black soil. This study addresses the challenges of modern agriculture by investigating the impact of straw incorporation on water soluble organic carbon (WSOC - I have proposed to change the abbreviation throughout the text). The paper is well written. Please consider using (DOM, DOC) , (dissolved organic matter, dissolved organic carbon) instead Water soluble organic carbon (WSOC).
• What does it add to the subject area compared with other published material? Potential to improve the fertility of weaker soils, which is important from an environmental and crop yield perspective.
• What specific improvements should the authors consider regarding the methodology? I have no comment. Is ok.
• Are the conclusions consistent with the evidence and arguments presented and do they address the main question posed? Please also explain why this is/is not the case. Yes, I have no comments
• Are the references appropriate? Yes.
• Any additional comments on the tables and figures . I have no comment.
Author Response
Comments1• What is the main question addressed by the research? This article is about Characteristics of Water soluble Organic Matter Structure in Planosol Profiles through Straw Incorporation: A Fluorescence Spectroscopy Study. This study demonstrates the effect of improving barrier soil and maintaining sustainable agriculture by enhancing soil quality, due to the addition of straw.
Response1: Thank you for raising this critical question. The primary objective of this study is to investigate how straw incorporation alters the structural characteristics of WSOC in Planosol profiles and to evaluate its potential role in improving the quality of barrier soils (e.g., nutrient-deficient layers beneath black soil) for sustainable agriculture. By combining fluorescence spectroscopy with soil chemistry analysis, we specifically address: This research question directly targets the knowledge gap in understanding organic matter dynamics in Albi-Boric Argosols subjected to straw management, a soil type critically understudied in the context of agricultural sustainability.
Comment 2 :Do you consider the topic original or relevant to the field? Does it address a specific gap in the field? Please also explain why this is/ is not the case. Yes, the topi cis original, but lanosol is characterized by a nutrient-deficient layer beneath black soil. This study addresses the challenges of modern agriculture by investigating the impact of straw incorporation on water soluble organic carbon (WSOC - I have proposed to change the abbreviation throughout the text). The paper is well written. Please consider using (DOM, DOC), (dissolved organic matter, dissolved organic carbon) instead Water soluble organic carbon (WSOC).
Response2: We sincerely appreciate the reviewer’s recognition of the topic’s originality and relevance. As rightly noted, Planosols are characterized by a nutrient-depleted subsurface layer beneath fertile horizons, posing significant changes to crop productivity. Factually, I couldn’t find any research focused on the DOC of straw incorporation in the Albi-boric argosols except our team now. Following the reviewer’s suggestion on terminology precision. We have replaced "WSOC" with "DOC" (dissolved organic carbon) in the revised manuscript. This enhances conceptual clarity and cross-study comparability.
Comment 3: What does it add to the subject area compared with other published material? Potential to improve the fertility of weaker soils, which is important from an environmental and crop yield perspective.
Response: Thank you for your positive comments and valuable suggestions to improve the quality of our manuscript. Please see below or line 479-491.Different feedback on straw application is observed due to the different soil types. Studies have shown that combined application of organic and inorganic fertilizers in black soil, red soil, paddy soil, and fluvo-aquic soil can enhance both DOC and total organic carbon content38, though differences exist between soil types. Red soi exhibits higher proportions of microbially derived components39, black soil shows varying degrees of increase in anmine substances35, while combined application of chemical fertilizers with straw returning in paddy soil leads to more complex and stable aromatic compound structures40. Different ferilization materials, straw return methods, and tillage practices have been found to increase the content of relatively simple-stuctured fulvic acid componts in DOC while simplifying their structural complexity12,41.
Thank you for your suggestions.
Reviewer 3 Report
Comments and Suggestions for Authors
attached

attached
Author Response
We appreciate the time and effort that you and the reviewers dedicated to providing feedback on our manuscript and we are grateful for the insightful comments and valuable improvements to our manuscript.
The respones has been uploaded as attachment.
If there have any further questions, please do not hesitate to communicate with me in a timely manner to discuss the manuscript.

Round 2
Reviewer 1 Report
Comments and Suggestions for Authors
I am satisfied with the changes made by the authors.
Reviewer 3 Report
Comments and Suggestions for Authors
accept in the present form